# Human Breast Milk: The Key Role in the Maturation of Immune, Gastrointestinal and Central Nervous Systems: A Narrative Review

**DOI:** 10.3390/diagnostics12092208

**Published:** 2022-09-12

**Authors:** Margarita Dimitroglou, Zoi Iliodromiti, Evangelos Christou, Paraskevi Volaki, Chrysa Petropoulou, Rozeta Sokou, Theodora Boutsikou, Nicoletta Iacovidou

**Affiliations:** Neonatal Department, Medical School, National and Kapodistrian University of Athens, Aretaieio Hospital, 11528 Athens, Greece

**Keywords:** breast milk, immune system, gastrointestinal system, central nervous system, neonate

## Abstract

Premature birth is a major cause of mortality and morbidity in the pediatric population. Because their immune, gastrointestinal and nervous systems are not fully developed, preterm infants (<37 weeks of gestation) and especially very preterm infants (VPIs, <32 weeks of gestation) are more prone to infectious diseases, tissue damage and future neurodevelopmental impairment. The aim of this narrative review is to report the immaturity of VPI systems and examine the role of Human Breast Milk (HBM) in their development and protection against infectious diseases, inflammation and tissue damage. For this purpose, we searched and synthesized the data from the existing literature published in the English language. Studies revealed the significance of HBM and indicate HBM as the best dietary choice for VPIs.

## 1. Introduction

Prematurity is the leading cause of death in children younger than 5 years old and a main reason for morbidity in the pediatric population. Annually, almost 15 million neonates are born prematurely (before 37 week of gestation), and the prevalence is increasing every year [1]. According to World Health Organization (WHO), among 184 countries, the percentage of preterm birth ranges between 5% and 18% [1]. In Greece, 4.7% of total births were premature between the years 1980 and 2008, with the percentage of prematurity reaching 9.6% in 2008 [2]. Very preterm neonates (<32 weeks of gestation) and extremely preterm neonates (<28 weeks of gestation) account for about 10% and 5%, respectively, of all premature neonates, with their morbidity and mortality being inversely related with gestation age [3,4].

In the past, respiratory distress syndrome was a major cause of morbidity and mortality for prematurely born neonates. From 1968 to 1973, it was the underlying cause of 50–70% of deaths in preterm infants in the United States [5]. Over the last twenty years, neonatal management of respiratory distress has improved, and as a result, mortality has decreased [6]. The use of antenatal corticosteroids and more advanced ventilation techniques have significantly decreased the incidence of respiratory distress syndrome [6]. A deeper knowledge of airway physiology and the angiogenesis and parenchymal damage of small airways has led to proper treatment and better outcomes. Neonatologists use more “sophisticated” ventilation methods and prefer early, less-invasive surfactant administration (LISA) strategies that reduce the need for mechanical ventilation and its complications, mainly in very preterm neonates [7,8,9]. 

Breastfeeding is the best nutrition for infants, especially for very preterm ones, since maternal milk is associated with a decreased risk for neonatal sepsis and NEC [10]. Human breast milk (HBM) factors such as immunoglobulins, lactoferrin, lysozyme, oligosaccharides, fatty acids, growth factors and anti-inflammatory factors are only some of the valuable components of HBM that induce the growth and development of the newborn organism.

This review examines the immaturity of VPI immune, gastrointestinal and neurological systems (Figure 1), which makes them more susceptible to diseases when compared to term and late and moderately premature infants (>32 weeks of gestation), and highlights the beneficial impact of HBM components on the maturation of these systems [11,12].

For this purpose, we conducted a narrative review of the existing literature. We searched the PubMEd database in English, and we also searched the citations within the articles. All types of research studies were included (e.g., systematic review, clinical trial, observational studies, animal studies). The key words we used were human or breast milk, very premature infants or neonates, immune system, gastrointestinal system, nervous system and neurodevelopment. 

## 2. Immune System Immaturity

Neonatal sepsis represents a critical clinical issue for neonatologists as neonates, especially VPIs, are at increased risk for severe infectious diseases. Recent studies report that 10–30% of VPIs develop late-onset sepsis at least once, and almost 18% of them finally die [45,46]. This vulnerability is mainly due to the immaturity of their physical barriers and to their innate and adaptive immunity systems not being fully developed [47]. Interestingly, VPIs are at greater risk for infectious diseases even one year after birth. According to an Austrian observational study, a negative correlation between duration of pregnancy in weeks and total number of infections during the first year of life was found (*r*(143) = −0.394, *p* < 0.001) [48]. 

### 2.1. Skin and Mucosal Barriers

Skin and mucosa are the first line of defense against pathogens. These physical barriers are not fully developed in premature infants. “Vernix caseosa”, a lipid-containing physical membrane that covers the skin and offers hydration, pH regulation and antimicrobial protection, begins to be produced during the third trimester [13,49]. “Stratum corneum”, the outer layer of epidermis, starts developing from the 15th week, and is fully developed around 34 weeks of gestation [14]. Therefore, neonates born before 30 weeks have inadequate production of “vernix caseosa” and thinner skin. 

Protection from mucosal barriers in VPIs is also impaired. Their intestinal mucosa is characterized by immature enteric cell junction and decreased production of protective factors, such as IgA immunoglobulins, which normally prevent the diffusion of enteric pathogens and toxins [50]. VPIs are also more vulnerable to respiratory pathogens since they have lower levels of local protective cells and expression of Toll-like receptors and reduced surfactant proteins produced by type II alveolar cells, antimicrobial proteins and peptides in their respiratory tract [47,51].

### 2.2. Innate and Adaptive Immunity

Innate cellular response and inflammatory response are also different in VPIs. The number of phagocytes is reduced, titers of cytokines and complements are lower, Th1-helper cell and macrophage activation is impaired, and opsonization and phagocytosis are less effective [15,52]. As a result, VPIs are more susceptible to infections by intracellular microorganisms. Impaired Th17-helper cell function in these neonates and lower levels of circulating antimicrobial proteins and peptides lead to vulnerability to extracellular pyogenic bacteria as well [16,17]. 

Azizia M. et al., reported that VPIs had significantly lower production of Tumor Necrosis Factor-a (TNF-a), and their monocytes expressed less Major Histocompatibility Complex (MHC) class II antigen after in vitro stimulation with lipopolysaccharides (LPS) [18]. Significantly lower production of proinflammatory cytokines (IFN-𝛾, TNF-𝛼 and IL-6) was also reported in septic VPIs compared with preterm infants. Specifically, the mean values of INF-γ, TNF-a and IL-6 in VPTN were 0.117 ± 0.31 pg/mL, 1.646 ± 1.12 pg/mL and 21.5 ± 22.80 pg/mL, respectively, while in preterm neonates, they were 3.095 ± 2.71 pg/mL, 4.995 ± 4.12 pg/mL and 866.64 ± 1233.11 pg/mL, respectively (*p* < 0.01, Student’s *t*-test) [19].

Adaptive immunity is responsible for any antigen-specific immune response (cellular or humoral). Maturation of adaptive immunity is acquired later in early childhood, so all newborns have immature adaptive immunity [15]. Premature infants have lower levels of circulating lymphocytes, contrarily to full-term ones [20]. According to a relative study, in VPIs, circulating lymphocytes remain relatively lower, even at the postnatal age of 12 months (total lymphocyte count (cells/mL) in VPIs and in term infants: 4180 (2411–6245) vs. 5902 (3882–9184), *p* < 0.001) [16]. Vertical transition of maternal antibodies via the placenta protects neonates after birth. Maternal IgG titers to fetal circulation remain low until the end of the second trimester of gestation, when they reach almost 10% of the maternal levels, and they finally exceed maternal levels at the end of pregnancy [21,53]. Hence, a preterm labor disrupts transplacental transmission of maternal IgG immunoglobulins, hence the significantly lower levels found in the serum of VPIs [22]. 

### 2.3. Immaturity of Gastrointestinal Tract

Necrotizing Enterocolitis (NEC) is a severe disease that mainly affects preterm neonates, and its incidence is inversely correlated with gestation age [54]. Up to 5% of these neonates and up to 10% of extremely preterm ones (born before 28 weeks of gestation) develop NEC, which has a mortality higher than 10% for these infants [55]. Infants that survive are also in risk of long-term gastrointestinal complications, such as short bowel syndrome with or without intestinal failure, cholestasis and liver disease, adhesions, dysmotility and feeding problems [56,57]. Immaturity of a VPI’s gastrointestinal (GI) tract in combination with a disturbance of normal enteric microbiome seems to contribute significantly to the pathophysiology of NEC [58,59]. 

Enteric epithelium consists of six main types of cells: enterocytes, which are the most abundant cells in intestinal epithelium and whose function is nutrient and water absorption; goblet cells, which produce mucus; hormone-secreting enteroendocrine cells; Paneth cells, which are found in the small intestine and secrete antimicrobial factors; tuft cells; and antigen-presenting M cells [13]. 

In full-term neonates, the external surfaces of highly polarized intestinal epithelial cells are tightly attached to each other, forming a protective barrier. In VPIs, this enteric barrier is impaired by increased permeability and decreased first-line defense against pathogens [24,25,26,60]. McElroy SJ et al. reported that neonates with NEC presented significantly lower levels of goblet and Paneth cells compared to controls (*p* < 0.001) [27]. In addition, intestinal mucus production by goblet cells, which consists of glycosylated proteins (mucins) and other antibacterial factors, is also decreased. The production of various antimicrobial factors by Paneth cells is decreased as well [23]. Finally, immaturity of the mechanism that regulates intestinal peristaltic motion, which normally prevents the junction of bacteria to the intestinal mucosa, has also been described in premature neonates [28,29]. 

Colonization of the gut microflora begins in utero through the amniotic fluid, but it mainly occurs postnatally [61,62]. The mode of delivery and infants’ mode of feeding significantly influence gut microbiome [63,64]. Long-term hospitalization in neonatal intensive care units and prolonged use of antibiotics, mainly in VPIs, also affect intestinal flora. Studies report that preterm infants in NICUs are colonized mainly by Proteobacteria and other bacterial species, such as Staphylococcus, Enterobacteriaceae, Enterococcus and Clostridia, and less by Bifidobacteria, which is a protective bacterium of the normal gut flora [30,31,32,33]. In a recent randomized trial, Reyman M et al. studied the impact of the prophylactic use of common antimicrobial factors during the first 48 h of life in the microbiome of infants born after the 36 weeks of gestation [34]. They reported that the use of antibiotics, even for a short period of time, led to decreased levels of several species of Bifidobacteria and increased levels of several species of Klebsiella and Enterococcus, and this alteration in gut microbiome remained until the first year of their life. Russell JT et al. found a negative correlation between antibiotic administration in preterm infants and Veillonella concentration [35]. Specifically, during antibiotic treatment, Veillonella was not detectable in stool, while its concentration before treatment was 3.23 × 10^6^ cells/gram (73%). Several species of Veillonella are Gram-negative bacteria that seem to increase titers of Gamma-aminobutyric acid (GABA) in intestinal mucosa. These neurotransmitters can normally be absorbed and transferred to the developing brain, where their role is essential. 

Toll-like receptors (TLRs) are a family of transmembrane proteins that recognize pathogens’ patterns and stimulate innate immune response. TLR4 recognizes lipopolysaccharide and lipoteichoic acids, patterns that are present on Gram-negative and -positive bacteria [65]. Additionally, TLR4 has an essential role in the development of the GI tract, which explains why, in premature neonates, TLR4 is expressed in high levels [36]. TLR4 in VPIs has been associated with intestinal damage and NEC. When TLR4 recognizes the specific patterns of enteric bacteria, they induce an immunological response, which finally leads to tissue inflammation, epithelium cell apoptosis and reduced local blood perfusion [36,66,67,68]. 

### 2.4. Immaturity of Cerebral Tissue 

VPIs are at greater risk for neurodevelopmental impairments. It is calculated that 6.2% of surviving VPIs have cerebral palsy, as well as a two to four times greater risk compared to full-term neonates to present with milder cognitive and behavioral problems [69,70,71,72,73].

The human brain is continuously developing during pregnancy, and many vital processes, such as neurogenesis, cellular proliferation, differentiation, myelinogenesis and synaptogenesis, occur until full term. Premature birth leads to more immature neurological tissue and decreased brain volume, and it is interesting that Nosarti C. et al. reported that this difference remains significant until adulthood [37]. The premature brain is in increased need for energy and nutrients for its function, growth and development [38]. Given the fact that extra-uterus life is more demanding than fetal life, and that VPIs usually have multiple health issues such as NEC, respiratory distress and sepsis, they are at higher risk for impaired brain nutrition and insufficient cellular development [74]. 

The blood–brain barrier (BBB) is well-developed from early fetal life, and its function is to protect the brain from various injurious substances in blood circulation [75]. The BBB is composed by endothelial cells, astrocyte end feet and pericytes [76]. Astrocyte end feet surround the cerebral vessels, and they probably provide structural stability. In the preterm neonatal BBB, fewer astrocyte end feet cover the cerebral vessels, which may contribute to the known susceptibility of preterm cerebral vessels to injury [39]. Additionally, systemic inflammation as a result of NEC or other diseases of prematurity could further disturb BBB permeability and contribute to tissue damage [77]. 

Ischemia or neuroinflammation of the premature brain disrupts neuronal and glial cell proliferation and differentiation and induces cellular apoptosis [72]. After the initial damage, a consecutive pathological tissue remodeling occurs [78,79], leading to further pathological neurological development and function [40,41].

GABA and glutamate are two essential neurotransmitters for appropriate development of neural networks, especially late in gestation. These neurotransmitters are structurally and functionally impaired in VPIs [42]. Basu SK et al. found a positive correlation between glutamine levels and post-menstrual age in VPIs (β/*p*-value, 0.22/0.02) [43]. In another study, preterm neonates also had lower levels of GABA and glutamate concentration in relation to term infants (*p* = 0.049 and 0.005, respectively) [44].

## 3. The Importance of HBM for Very Preterm Neonates

In the past, neonatologists used to delay enteral feeding to the VPI, sometimes up to the fourth day of life or longer. However, this strategy has been replaced by early introduction of HBM since recent data support that early enteral feeding with HBM is beneficial for the GI system [80] and for neurodevelopment and could possibly reduce the risk for NEC and sepsis (Figure 2) [81,82,83]. 

### 3.1. Immunology of HBM

HBM is essential for VPIs as it provides various factors, such as immunoglobulins, lactoferrin, lysozyme, G-CSF, oligosaccharides, lipids, cytokines and growth factors. These nutrients directly protect newborns (passive immunity), and additionally, they modulate the immune response inducing the maturation of their immune system and moderate inflammation responses (anti-inflammatory function) [84]. In a relative study, Patel AL et al. found that increasing the average daily dose of HM from day 1 to 28 in VLBW infants (mean gestation age 28.1 ± 2.4 weeks) was associated with lower odds of sepsis (OR 0.981, 95% CI 0.967–0.995, *p* = 0.008) [129].

Immunoglobulins are abundant in human breast milk. The major isotype is secretory IgA (80–90%), followed by IgM (8%) and IgG (2%) [130]. Secretory IgA is a dimeric of two IgAs linked together with a joining chain and a secretory component [131]. This domain remains stable in the preterm stomach even 3 h after feeding, is attached to the neonatal intestinal mucus, binds to local microorganisms and toxins, and blocks their penetration into the epithelium [85,86,132]. Secretory IgA is higher in colostrum [87,88], and titers are even higher in very preterm HBM (*p* < 0.05–0.0001) [89].

Lactoferrin is an iron-binding glycoprotein, directly protecting newborns against enteric pathogens. Through iron binding, lactoferrin blocks its use by pathogens that need iron for their survival and multiplication. A recent multicenter clinical trial in Peru showed that lactoferrin supplementation decreased the risk of infections in low-birth-weight infants (LBW < 2500 g) and especially in those with very low birth weight (VLBW < 1500 g) (LBW: lactoferrin groups vs. placebo, 12.6 vs. 22.1%; VLBW: lactoferrin group vs. placebo, 20 vs. 37.5%) [90]. Another Cochrane meta-analysis found that lactoferrin supplementation reduces relative risk for late-onset sepsis and NEC in VPIs (RR = 0.59, 95% CI 0.40–0.87 and RR = 0.4, 95% CI 0.18–0.86, respectively) [91]. These data implicate a further protective action of natural lactoferrin for VPIs [92]. 

Lysozyme is an active enzyme found in extremely high concentrations in HBM. It lyses Gram-positive bacteria by deconstruction of proteoglycans at the cell surface and seems to have a synergistic function with lactoferrin against Gram-negative bacteria. After lactoferrin binding through LPS to Gram-negative bacteria, it creates pores through which lysozyme can insert into inner membrane and digest the proteoglycans [93]. The HBM of mothers of VPIs contains lysozyme in higher concentrations (*p* < 0.05–0.0001) [89].

Lactadherin is an HBM protein that binds to all human rotavirus strains, thus preventing infection by it [133]. This virus is quite common in VPIs. This could be explained by the lower concentrations of lactadherin in the milk of their mothers in contrast to that of mothers of full-term infants (*p* < 0.05–0.0001) [89].

HBM lipids are known for their nutritional value and for their protective role against infectious diseases. Salivary and gastric lipases break down lipids to free fatty acids and monoglycerides [94]. Long-chain polyunsaturated fatty acids (LCPUFAs) have significant antibacterial and antiviral action by acting directly against microorganisms or indirectly by producing bioactive molecules that enhance the phagocytic action of immune cells [134]. Finally, components that are found in the HBM lipid globule membrane possibly block the binding of pathogens to GI mucosa [95].

Colostrum and mature HBM are abundant in oligosaccharides. HBM oligosaccharides (HMOs) are produced by the mammary gland and have great immunomodulatory action [96]. They are resistant to digestion and are not absorbed in great amounts. They alter the immune response, degrading inflammation by interacting with intestinal lymphoid tissues or by interacting systemically with selectins, dendritic cells (DCs), integrins, and Toll-like receptors (TLRs) [97]. HMOs also improve the immune system response. Azagra-Boronat et al., in an animal study, reported that daily supplementation of 2′ Fucosyllactose during the first 2 weeks of life in suckling rats increased serum release of IgG (control vs. 2′FL group 3693.28 ± 180.15 vs. 5732.33 ± 284.11, *p* < 0.05) and IgA (control vs. 2′FL group 63.32 ± 1.4990.15 ± 3.37, *p* < 0.05) and increased T cell production in the lymph nodes of the mesentery (controls vs. 2′FL group 62.01 ± 2.12 vs. 74.97 ± 2.03, *p* < 0.05) [135].

Although the main role of HBM immunoglobulins is to provide passive immunity to newborns, they also modulate the adaptive immune response of breastfed infants. Maternal antibodies interact with host B and T cells and influence their programming [136]. In animal studies, lactoferrin was reported to affect immune response. Specifically, almost two-fold titers of IgG, IL-10 and TNF-a (*p* < 0.05) and increased numbers of natural killer cells (*p* < 0.01) were detected in piglets fed with formula milk supplemented with lactoferrin than in the control group [98,137]. Lysozyme also has anti-inflammatory action [84]. 

Cytokines and growth factors affect immune response by binding to intracellular receptors in target cells. Interleukin-10 (IL-10), transformation growth factor beta (TGF-β) and tumor necrosis factor receptor I and II (TNFR I, II) are some of the molecules that act against inflammation [132].

Breast milk cytokines, such as IL-7, also seem to induce the growth of the thymic gland; therefore, they enhance the immunity of neonates [99,100]. Thymic development and activity are better in breastfed infants than in formula-fed ones, and T cells are found in higher titers in their serum [101].

MicroRNAs (miRNAs) are tiny RNA fragments found in great amounts in HBM and can modulate genetic expression. Lately, there has been an increased interest in their possible protective effects for neonates. It seems that miRNAs interfere with the maturation and modulation of the immune system. MicroRNAs affect B and T cell differentiation and induce monocyte development and granulocyte reproduction [102,103]. 

Soluble TLRs and CD14 are present in HBM. They seem to interact with the TLRs of intestinal mucosa and prevent inflammatory response in VPIs’ gut [138]. 

Breast milk lipids, such as omega-3 polyunsaturated fatty acids (PUFAs), interfere with gene expression and immune cell migration and alter gut microbes, protecting neonates against inflammation. PUFAs affect Th1 and Th2 cell responses and alter the cellular membrane domain, increasing arachidonic acid [139]. 

### 3.2. HBM and Gastrointestinal System

The gastrointestinal tract of VPIs is predisposed to inflammatory damage and NEC. Feeding with HBM decreases the risk for NEC, in contrast to formula, probably due to the anti-inflammatory agents that it contains, some of which have already been described. Interestingly, in a relative study with 797 very low-birth-weight neonates (VLBW < 1500 g) with mean gestation age 28.4 ± 2.6 weeks, those with HBM feeding for ≥90% of their hospitalization had a significantly lower risk for death or NEC in contrast with neonates that did not receive human milk (mortality rate 7.9% vs. 0.0%, *p* = 0.016; NEC rate 10.5% vs. 0.0%, *p* = 0.005) [140]. In another observational study, neonates born before 33 weeks of gestation and fed exclusively with HM had a significantly lower risk for NEC onset after day 7 than controls (1% vs. 3.4%, respectively, *p* = 0.009) [141]. 

HBM also promotes GI barrier maturation and alters microbial colonization. Human milk oligosaccharides contribute significantly to it [104]. The most abundant oligosaccharide in HBM is HMO-2′-fucosyllactose (2′FL) [142]. Oligosaccharides are selectively consumed by Bifidobacterium species, which are the dominant microorganisms in full-term neonates’ gut flora. The GI microbiota of VPIs, however, differs, and Bifidobacteria are present in significantly lower levels, something that is reasonable as VPIs are usually hospitalized and are treated with antibiotics for long periods [143]. On the other hand, the preterm GI tract is mainly colonized by potentially pathogenic bacteria, such as Gram-negative Enterobacteriaceae of the Proteobacteria phylum [143,144]. Through breastfeeding, VPIs are nourished with HBM oligosaccharides and other nutrients, which promote the development of Bifidobacteria against enteric pathogenicity [142,145]. Lactoferrin also promotes the growth of protective bacteria against pathogens [105]. 

HBM growth factors also contribute significantly to the protection of VPIs’ gastrointestinal systems. They induce the maturation of the GI barrier and the production of mucus by goblet cells, making VPIs less vulnerable to GI damage and infection [106]. More specifically, epidermal growth factor (EGF), which is found in greater amounts in colostrum and in preterm HBM, enables the recovery of intestinal mucosa from damage. It is resistant to digestion, stimulates enterocyte proliferation and blocks programmed cell death, thus healing injured tissues [107]. Heparin-binding EGF-like growth factor (HB-EGF) is found in HBM in lower amounts and is the main growth factor contributing to the remediation of intestinal tissue. Radulescu A et. al., in an animal study, reported that HB-EGF increases collagen deposition and enhances angiogenesis [108]. HB-EGF induces the maturation of the enteric nervous system and protects it from damage by NEC [109]. Hepatocyte growth factor (HGR) mainly induces organogenesis and maturation of the gastrointestinal system. It also interacts with vascular endothelial growth factor (VEGF) to promote angiogenesis [110]. Brain-derived neurotrophic factor (BDNF) and glial cell-line derived neurotrophic factor (GDNF) promote the maturation of the enteric nervous system and are contained in HBM [111]. 

Platelet-activating factor acetylhydrolase (PAF-AH) is detected in HBM and seems to have a protective role. It is an enzyme secreted by milk macrophages, and its biological action is to degrade platelet-activating factor (PAF), which can damage intestinal mucosa [112]. Another factor that possibly has a protective action against tissue damage and NEC is IL-8, a cytokine detectable in significant titers in HBM and which is stable in digestion. Its receptors CXCR1 and CXCR2 are expressed in the fetal GI tract early in gestation. Maheshwari A et al. reported that human fetal and adult intestinal cells, after in vitro administration of rhIL-8, presented augmented migration, proliferation and differentiation (*p* < 0.04) [113]. 

On the other hand, human formula contains nutrients that could induce tissue damage in VPIs. In a relative animal study by Singh P. et al., maltodextrin, a polysaccharide contained in human formulas, was correlated with increased injury scores [146].

Furthermore, progressively increased feeding of VPIs with HBM leads to sooner disengagement from parenteral nutrition, which indirectly decreases the risk of correlated complications, such as liver disease and central line-associated bloodstream infections (CLASBIs) [147,148]. 

### 3.3. Significance of HBM for Cerebral Tissue

HBM has a great role in the development of the neonatal brain. Vohr et al. reported that HBM is beneficial for extremely low-birth-weight neonates (ELBW < 1000 g) when it comes to neurodevelopmental outcome at 18 and 30 months corrected age [149,150]. A recent study showed that dominant ingestion of HBM (≥50% of total enteric intake) during the first 28 days of VPI life was correlated with better cognitive development at the age of 7 (IQ, mathematics and working memory test were 0.5 points higher for each additional day of dominant ingestion of HM (95% CI 0.1–0.2, 0.8–0.9)) [151]. 

VPIs require greater energy for physical growth and brain development compared to fetuses because they depend on the extra-uterus environment [38]. This means that it is essential for VPIs to receive all necessary macro- and micronutrients and energy via their nutrition. Formula and donor milk are alternative choices; however, enteral feeding with HBM is preferred, as it has been associated with better long-term neurodevelopmental outcomes [152]. 

Lipids account for more than 50% of cerebral tissue’s dry weight and are the nutrients that have mostly been related to brain development. Docosahexaenoic acid (DHA) and arachidonic acid (AA) are two LCPUFAs that are involved in many functions of neural cells, such as neurogenesis, neuronal migration and synaptogenesis [114,115]. Their role is also crucial for physiological retinal development [116]. Girls fed with HBM rich in DHA had better school performance that non-breastfed ones (β = 2.96 points, 95% CI 0.24; 5.69) [117], whereas such a correlation was not found in boys. The possible benefit of supplementation with PUFAs in VPIs and in breastfeeding mothers is a controversial topic. However, two recent Cochrane systematical reviews reported that there is no beneficial effect on neurodevelopment [118,119]. 

Sphingomyelin is a phosphosphingolipid that constitutes an essential molecule of the myelin sheath and has higher levels in HBM than in formulas. Infants fed with products abundant in sphingomyelin during the first 3 months postnatally had better verbal development at 2 years of life and greater myelination of the brain. Increased proliferation, maturation and differentiation of oligodendrocyte predecessor cells and increased production of myelin after in vitro sphingomyelin supplementation were observed [120]. 

Gangliosides are glycosphingolipids that are present in HBM in higher levels than in formulas [121]. They are essential in synaptogenesis, in neurotransmission, in neurogenesis, in neuronal maturation and in memory formation [122,123]. 

Human milk oligosaccharides are the third most abundant nutrient in HBM after lipids and lactose and possibly stimulate neurodevelopment [153]. Berger PK et al. showed in an animal study that chronic oral administration of the oligosaccharide 2′-fucosyllactose (2-FL) to mice and rats led to better performance in behavior tests and to increased expression of molecules that are involved in memory function. Yet, in human neonates, the correlation between 2-FL and cognitive development was found only in the first month of life but not at six months of life [124].

HBM contains various micronutrients, such as vitamin B6 (pyridoxal) and carotenoids. These nutrients seem to be of great significance for developing cerebral tissue. Pyridoxal is a water-soluble vitamin that is involved in the synthesis of GABA, dopamine and serotonin. Boylan LM et al. observed a positive correlation between the levels of B6 in breast milk and infant scores on habituation (r = 0.94, *p* < 0.05) and autonomic stability scales (r.34, *p* < 0.05) of NBAS at 8–10 days after birth [125]. Carotenoids are natural pigments that are normally contained in fruits and vegetables. According to a relative study, infants fed with milk that contained greater carotenoid levels (b-carotene and lycopene) presented greater psychomotor development at the first and third month postnatally (β = 0.359, *p* ≤ 0.05) [126]. 

Lactoferrin’s role in neurodevelopment has not yet been clarified; however, it seems that it could have a beneficial function in the neonatal brain. In some animal studies, supplementation of lactoferrin to subjects led to greater neurodevelopment results [127,128].

Although HBM contains many components that could lead to better neurodevelopmental outcomes, their supplementation to formula milk has not yet been proven beneficial. According to a hypothesis, the better results in neurodevelopment of breastfeeding VPIs could be attributed, not to the nutrients of HBM, but to the maternal–infant interaction. Mothers that feed their infants with their HBM seem to allow more hours interacting with their infant, which could lead to a better-developing brain. This hypothesis is supported by data that do not find donor milk beneficial for neurodevelopment [154].

Finally, HBM leads to a decreased risk for neonatal sepsis, NEC and neonatal inflammation. By protecting VPIs from these clinical conditions, HBM seems to have an additional, indirect neuroprotective effect. 

## 4. HBM Fortification

Recent data indicate that HBM should be preferred instead of formula for VPIs. However, ELBW and VLBW neonates need, for their adequate growth and neurodevelopment, higher levels of protein and minerals (mainly calcium and phosphate) than those found in HBM at usual feeding volumes [155]. Additionally, maternal factors such as nutrition, body mass index, period of lactation and ethnicity can affect HBM composition [156]. For this reason, it is essential that HBM is fortified with necessary nutrients in neonatal intensive care units. 

## 5. Conclusions

VPΙs are a high-risk population for neonatologists and pediatrics. Their immature immune and gastrointestinal systems make them more vulnerable to perinatal sepsis, intestinal inflammation and mucosal injury. Additionally, their cerebral tissue is not fully developed and is at even higher risk of secondary damage. HBM’s nutrients protect VPΙs both directly and indirectly by degrading inflammation and enhancing tissue regeneration and development. Immunoglobulins, lactoferrin, lysozyme, lactadherin and lipids are some of the most important HBM nutrients that provide passive immunity to VPIs. Additionally, HMOs, proteins, cytokines, GFs, MiRNAs, TLRs and CD4 modulate the immune response and protect VPIs from inflammation damage. Finally, factors such as BDNF, GDNF, HMOs, lipids and micronutrients contribute to maturation of the gastrointestinal and nervous systems. When maternal milk is unavailable, donor HBM could be an alternative choice [156]. 

Although HBM contains numerous valuable nutrients, it should be fortified with protein and minerals for the adequate growth and development of VLBW neonates. Interestingly, maternal–infant interaction during breastfeeding may also contribute to the development of VPI cerebral tissue. Further research is needed in order to better understand the way that HBM leads to neurodevelopment. 

## Figures and Tables

**Figure 1 diagnostics-12-02208-f001:**
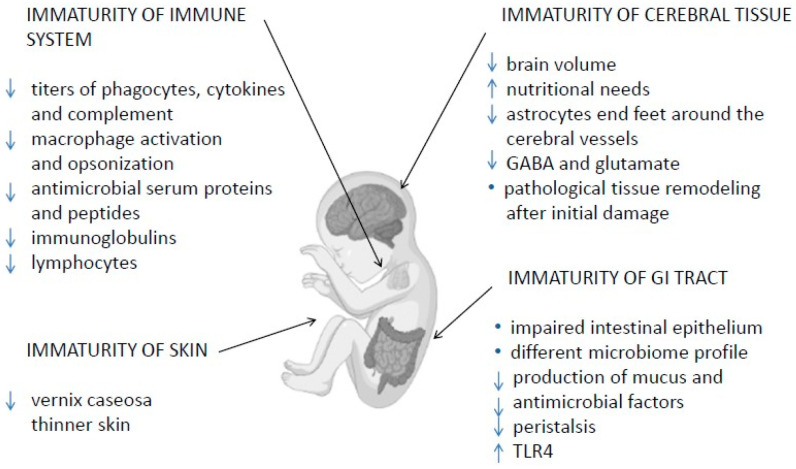
Immaturity of immune, gastrointestinal and nervous systems in very preterm neonates [13,14,15,16,17,18,19,20,21,22,23,24,25,26,27,28,29,30,31,32,33,34,35,36,37,38,39,40,41,42,43,44].

**Figure 2 diagnostics-12-02208-f002:**
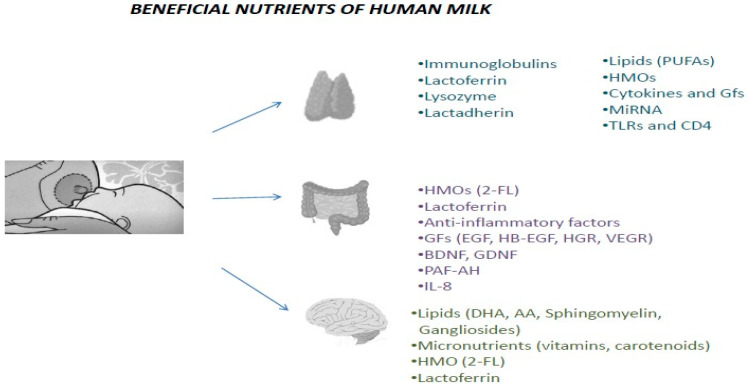
Nutrients of HBM that provide passive immunity, have anti-inflammatory action and induce the maturation of immune, gastrointestinal and nervous systems in VPIs [83,84,85,86,87,88,89,90,91,92,93,94,95,96,97,98,99,100,101,102,103,104,105,106,107,108,109,110,111,112,113,114,115,116,117,118,119,120,121,122,123,124,125,126,127,128].

## Data Availability

All data generated or analyzed during this study are included in this article. Further inquiries can be directed to the corresponding author.

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
