# Peer review of "Human Breast Milk: The Key Role in the Maturation of Immune, Gastrointestinal and Central Nervous Systems: A Narrative Review"

_diagnostics, 2022, doi:10.3390/diagnostics12092208_

Round 1

Reviewer 1 Report

This is a comprehensive review about the immaturity of very preterm neonates’ immune, gastrointestinal and neurological system and about the importance of human milk.

There is certainly a lack of statistical data on studies in the literature that highlight the differences, in terms of RR or OR with significance level (p-value or confidence intervals), between preterm neonates fed with human milk vs. preterm neonates fed with formula milk. This is essential to better understand the concepts that the authors reported in the THE IMPORTANCE OF HM FOR VERY PRETERM NEONATES section. Some subsections describe the consequences for the preterm infants of a lack of factors like immunoglobulins, lactoferrin, lysozyme, lactadherin, oligosaccharides, lipids, proteins cytokines and growth factors, but never expressed in quantitative indicators, nor in terms of time: in the first year of life or later?

Some minor revisions:

line 31 of the abstract "...protectS the neonate...".

line 156 put the full stop.

lines 166-167 specify entirely the word "spp.".

lines 481-482 of the ABBREVIATIONS: put "GABA" and G-CSF in correct alphabetical order.

line 512 reference 4: I don't understand what the initial letters are. If they refer to the authors, write them down in full.

Reviewer 2 Report

The purpose of the review is not made clear why this review should be considered for a novel publication. There are multiple reviews, especially narratives, that explore the immune, gastro and neuro system of very preterm infants focus on the breast milk components:

PMID: 23178065 (2014) / PMID: 30386758 (2018) / PMID: 31185620 (2019) / PMID: 33262207 (2020) / PMID: 32092925 (2020) / PMID: 33643310 (2021) / PMID: 33245565 (2021) / PMID: 34207732 (2021) / PMID: 34030819 (2021)

Perhaps the authors would like to focus the review on a more specific field such as milk immune cells and perform a systematic review or even a meta-analysis.

Among others, the abstract is rather poor, it should go deeper into the points discussed on the narrative review. 

Introduction: Defines very preterm neonate (line 51). It reports data in relation to % (line 52-54, 62), on the other hand, the authors refer to BPD? the current definition does not consider pneumonia in its diagnosis. Neurodevelopmental events are not considered respiratory distress events (line 54). The relationship between respiratory distress and ventilatory mechanics along with surfactants should be introduced.

Author Response

Manuscript ID: diagnostics-1861258

Journal: Diagnostics

Manuscript Title: HUMAN BREAST MILK: The key role in the maturation of immune, gastrointestinal and central nervous system: A narrative review

Letter of Response to Reviewer 2

Date: 27/08/2022

Dear Editor in Chief, Prof. Dr. Andreas Kjaer and Ms. Carol Zhang

Thank you for giving us the opportunity to submit a revised draft of our manuscript entitled “HUMAN BREAST MILK: The key role in the maturation of immune, gastrointestinal and central nervous system: A narrative review” for publication in the Diagnostics.

We appreciate the time and effort that you, the reviewers dedicated to providing feedback on our manuscript and we are grateful for the insightful comments on our paper. We have addressed all the comments made by the reviewers. Please see below, for a point-by-point response to the reviewer’s 2 comments and concerns. All changes that we have made, after taking into consideration the journal’s manuscript style recommendations and the reviewer’s comments, are incorporated in this letter and in the revised manuscript text, Figures and Tables, as well, in yellow color.

Point-by-point response to Reviewer 2

Reviewer 2

The purpose of the review is not made clear why this review should be considered for a novel publication. There are multiple reviews, especially narratives, that explore the immune, gastro and neuro system of very preterm infants focus on the breast milk components:

PMID: 23178065 (2014) / PMID: 30386758 (2018) / PMID: 31185620 (2019) / PMID: 33262207 (2020) / PMID: 32092925 (2020) / PMID: 33643310 (2021) / PMID: 33245565 (2021) / PMID: 34207732 (2021) / PMID: 34030819 (2021)

Perhaps the authors would like to focus the review on a more specific field such as milk immune cells and perform a systematic review or even a meta-analysis.

We agree that most narrative reviews in contrast with systematic reviews or meta-analyses cannot be considered 100% novel. However, they are more focused on points considered significant by the authors something that is not possible in systematic reviews.

At the present, the goal of this manuscript is not the writing of a systematic review or a meta-analysis. This is a very good idea for us in the near future.

We have reviewed the relevant literature as well as the manuscripts referred by the reviewer. Most manuscripts are more focused on either a specific system or on the beneficial effects of HBM. Therefore, we believe that our review is more comprehensive and will be interesting for the reader.

Among others, the abstract is rather poor, it should go deeper into the points discussed on the narrative review. 

We agree with the reviewer that the initial abstract was brief. Therefore, we made extensive additions in the abstract, which now discuss main points of the review and contain its purpose.

Introduction: Defines very preterm neonate (line 51).

Very preterm neonates are now defined.

It reports data in relation to % (line 52-54, 62), on the other hand, the authors refer to BPD? the current definition does not consider pneumonia in its diagnosis. Neurodevelopmental events are not considered respiratory distress events (line 54). 

We thank a lot the reviewer for the comment. Prematurity is a main reason of respiratory distress, neurodevelopmental disorders in later life and in some cases of BPD. It was not easily comprehensible so we made some changes in the original manuscript for better reading.

The relationship between respiratory distress and ventilatory mechanics along with surfactants should be introduced.

We agree with the comment. We explained more detailed the improved outcome of the respiratory distress of neonates because of the less invasive modes of ventilation the use of surfactant and antenatal/postnatal corticosteroids.  

Reviewer 3 Report

The narrative review “THE IMPORTANCE OF HUMAN MILK FOR VERY PRETERM NEONATE…” covers several aspects of very preterm neonates and the roles of human milk as a biological protective agent in their growth and development.

The very preterm is a missing event of the third trimester, which causes premature babies since fetal brain development and its maturity occur during that tenure.

The review has essentially two important components. First, very preterm and its pathogenesis. Second, is the importance of human milk for these subjects. The pathogenesis of prematurity which included immaturity of the immune, gastro and nervous system, is somewhat established. To make it interesting, the author may add information on the possible risks of preterm in their later life.

The abstract is the summary of a work. It is poorly complied. I did not find the purpose of the review in the abstract.

The title is a bit confusing and should be made precise.

The authors used terms like “maternal milk” or “human milk” in the manuscript. It is important to apply uniformity like “human breast milk” as the articles emphasized in humans.

The rationale to include very preterm is not included in the objective. Extreme preterm, very preterm, preterm and their prevalence globally and locally should be added in the introduction.

The strategy, and methodology to construct this narrative review are not mentioned.

The manuscript has a structural problem with figures and headings. There are too many subheadings with a short description of a topic, especially in the latter half (page 9 onward) of the manuscript.

Unlike a book chapter, a narrative review does not require subheadings like lipid, miRNA, TLR etc.

Fig. 2 needs to revise with specific changes to these parameters preferably with proper references.

The importance of human milk should be viewed as therapeutic for very preterm babies. Thus, authors should discuss potential functional changes of human milk that can improve the growth and development of very preterm neonates.

The conclusion should highlight the key message of the article in detail.

The micronutrient content varies in human milk due to maternal store, race, food intake and many more. One needs to consider those factors in targeted human milk fortification for preterm infants.

Typo

156 line--postnatally

193- spacing between that this

223- neurodevelopment

274- sentence framing

391- that spelling

401- neurotransmission

428- neurodevelopmental

Round 2

Reviewer 2 Report

We thank the authors for this revision of the manuscript. The abstract, along with the purpose and some concepts, have been clarified. However, some considerations remain to be reinforced and I would like the authors to have the opportunity to comment on them.

The authors comment that "narrative reviews focus on the points considered significant by the authors, something that is not possible in systematic reviews". In my opinion, I do not believe that this is true for systematic reviews, which consider a more exhaustive methodology in the search for information and organize the results generated by other studies than narratives. Perhaps, this review could be considered systematic if it follows the PRISMA criteria.

On the other hand, the review presented for consideration contemplates several specific systems of the neonate, such as gastrointestinal and immune systems, and they are also related to the benefits of MBH. I am surprised that references such as: PMID: 31185620 or PMID: 33262207, nor original data from animal models (PMID: 32753526, PMID: 33445698, PMID: 26172126) had been not included.

Round 3

Reviewer 2 Report

No further comments